# The Mediating Role of Resistance to Innovative Technology between the Characteristics of Innovative Technology and Sustainable Use of Innovative Payment Service

**Jaepil Yoo**

Department of Management Engineering, Sangmyung University, Seoul 03016, Korea; bond1919@naver.com

**Abstract:** The purpose of this study is to verify the structural relationship between innovative technology characteristics (recognized usefulness, ease of use, perceived risk), resistance to innovative technologies and acceptance intentions in order for unmanned order payment services to become a sustainable industry. A survey was conducted on experienced users of unmanned order payment services residing in Seoul, and the main analysis results are as follows: first, after verifying the effect of innovative characteristics of unmanned order payment services on resistance to innovative technology, the perceived usefulness and perceived ease of use of unmanned order payment services negatively affect resistance to innovative technology, and perceived risk has a significant positive effect on resistance to innovative technology. Second, after verifying the effect of resistance to innovative technology to unmanned order payment services on acceptance intention, consumers' resistance to unmanned order payment services negatively affects acceptance intention. Third, verifying the effect of characteristics of innovative technology of unmanned order payment services on acceptance intention, the perceived usefulness of unmanned order payment services directly had a positive effect on acceptance intention, but the perceived ease of use and perceived risk. Fourth, verification of the mediating effect of resistance to innovative technology in the relationship between the characteristics of innovative technology of unmanned order payment services and the acceptance intention proved significant mediating effects of all of perceived usefulness, perceived ease of use, and perceived risk.

**Keywords:** sustainable use; unmanned order payment services; characteristics of innovative technology; technology management; sustainable development

## 1. Introduction

In the era of the Fourth Industrial Revolution, new payment technologies have been developed, but the technologies are limited in sustainable use by resistance to these innovative payment technologies, and efforts are required to continue their sustainable use. Recently, unmanned order payment systems have been distributed mainly to restaurants, coffee shops, and fast food outlets. It is expected that such unmanned order payment systems will be widely distributed to other stores. The positive aspects of such unmanned order payment systems such as convenience coexist with negative aspects such as accident risks. These consumer objections often prevent innovative technologies from providing sustainable services. When innovative technologies or services are introduced following modernization and futurization, the phenomenon of consumers' acceptance and the phenomenon of consumers' rejection exist simultaneously. Since consumers face psychologically very complex situations in the stage of accepting innovative technologies or services, attention should be also paid to the perspective of resistance [1,2]. Innovation requires consumers' changes, and resistance to sudden changes can be said to be a natural response of consumers. There are limitations in studies in the case of the innovative technology acceptance model (TAM) and the innovation diffusion model because these models approach only the positive aspects (adoption and technology diffusion) of innovation. However, it is

important to identify consumer resistance to become a sustainable innovation technology. Therefore, early innovation studies may be more meaningful if they are conducted with large numbers of innovation resisters rather than being conducted on diffusion with small numbers of innovation accepters [1,3]. In addition, it is pointed out that users' resistance to innovative technology is overlooked in adoption and diffusion studies despite that it is an important element [3].

In the process of adopting innovation, not only the characteristics of innovation but also the characteristics of consumers must be considered. Therefore, studies that consider the psychological characteristics of consumers are necessary [1,4]. This means that innovation is not a concept that is accepted because it is recognized to be useful to all consumers, and that resistance to innovation exists. Users' resistance to innovative technology is not a concept opposite to acceptance or negative concept but can be said to be a process that occurs in the process of accepting innovative technology [5,6]. Users' resistance to innovative technology can be said to be a concept referring to the state where consumers, who are in a state of psychological balance, refuse changes when the innovative technology or service is first introduced. The concept of users' resistance to innovative technology was first argued by Sheth [3]. Thereafter, Ram [1] argued that resistance such as a sense of threat felt due to changes because of the psychological state of consumers who refuse changes from the current state occurs in the process of accepting innovation while organizing the concept of user's resistance to innovative technology.

Thus far, studies that applied the TAM have been conducted in various industrial fields. First, there is a study that applied the TAM to the factors that affect farms' acceptance of ICT convergence technology in order to propose alternatives and solutions for ICT convergence expansion and sustainable settlement in the rural sector [7]. In the field of the tourist hotel industry, a study that used the TAM to determine the acceptance attitude of consumers for the sale of hotel products using social commerce platforms was conducted [8]. In the field of the IT industry, which is closely related to cutting-edge technologies, there are studies that applied the TAM to various technologies and new products such as a study that applied the expanded TAM to artificial intelligence (AI) speaker models that are supplied to the market in earnest these days for analysis in relation to use intentions [9] and a study that applied the TAM to the smart wearable market represented by Apple Watch and Galaxy Watch [10]. As such, studies that used the TAM have been conducted with various fields, industries, technologies, and products, but there has been no study that applied the TAM to unmanned order payment systems thus far. Therefore, the purpose of this study is to analyze the perceived usefulness, perceived ease of use, and perceived risk of unmanned order payment technology by applying TAM, a representative innovation technology in the era of the 4th Industrial Revolution, to become a sustainable industry. Through the foregoing, this study is intended to derive the characteristic factors of the innovative technology of unmanned order payment systems that affect consumers' resistance to innovative technology and technology acceptance for unmanned order payment systems.

The specific purpose of this study is as follows.

First, to verify the effects of characteristics (perceived usefulness, perceived ease of use, and perceived risk) of unmanned order payment technology on resistance to innovative technology.

Second, to verify the effects of resistance to innovative technology on acceptance intention of unmanned order payment systems.

Third, to verify the effects of characteristics (perceived usefulness, perceived ease of use, and perceived risk) of unmanned order payment technology on acceptance intention of unmanned order payment systems.

Fourth, to verify the mediating effects of resistance to innovative technology in relationship between characteristics of unmanned order payment technology and acceptance intention.

This study consists of a total of six sections. Section 1 presents the necessity and purpose of this study. Section 2 is the theoretical background part, and deals with the concept

of TAM, innovative technology resistance and acceptance intention, and previous studies. Section 3 presents the research model and hypothesis of this study, the operational definition of the research variable, and the statistical data processing method. Section 4 presents the results of empirical analysis, and Section 5 presents the results of this study. Lastly, Section 6 presents the limitations of this study and suggestions for subsequent studies.

## 2. Theoretical Background

### 2.1. Technology Acceptance Model (TAM)

The technology acceptance model (TAM) was developed as a theoretical framework to figure out what are the factors that affect members' acceptance of information technologies introduced to improve organization members' performance based on Ajen and Fishbein's theory of reasoned action (TRA), which is a behavioral intention model to predict future behavior while looking at the current situation [11]. The TAM, which was started and developed for such purposes, mainly focuses on grasping the causal relationships intertwined among the beliefs, positive attitudes, negative attitudes, intentions to use, and actual behaviors held by members toward specific innovations and finding external factors that affect the acceptance process [12]. In the case of Davis's [13] technology acceptance model (TAM), shortcomings of being too simple and emphasizing only users' judgment about technology have been pointed out [14]. In addition, although the TAM has been used extensively to explain users' technology acceptance, since the conceptualization of users' beliefs and attitudes only shows external motives, the need to expand the TAM to include information technology users' internal motives has been raised in some studies [14]. Therefore, external variables were added to the technology acceptance model to present the expanded technology acceptance model [15].

Perceived usefulness is also an important factor in explaining consumers' acceptance of technology in TAM, but is considered the same concept as the perceived relative advantage mentioned in the innovation resistance model [16]. Perceived usefulness is one of the innovation characteristics that affect innovation resistance and acceptance in the innovation resistance model. Perceived usefulness is defined as the degree to which potential consumers perceive new innovation technologies as better than existing technologies that perform the same or similar functions, and perceived ease of use means ease of perception in using new innovation technologies. In addition, perceived risk refers to the degree of risk perceived by consumers even if there is no actual risk [17]. In the study of Lee et al. (2012), through empirical research, they found the relationship that the higher the user perceives the usefulness, the lower the innovation resistance is [18]. Previous studies have empirically suggested that new technologies are more useful when they show superior performance and value than existing technologies, and the faster users acquire technologies, the faster they settle in the market [19–22]. These findings imply that the greater the perceived usefulness and perceived ease of use, the higher the user's acceptance intention.

### 2.2. Resistance to Innovative Technology

Innovative technologies induce resistance to changes because they require changes to users [1]. Therefore, before interested in the acceptance and spread of innovative technologies, resistance to innovative technologies must be overcome before it can become a sustainable industry. Ram [1] presented a resistance to innovative technology model for the first time [1]. Sheth [3], who conducted a study on the concept of resistance to innovative technology for the first time, presented the concept of resistance in acceptance, not the resistance as a concept opposite to innovativeness and he stated that the negative feeling caused by innovation can be expressed as uncertain emotion toward innovative technologies, lack of trust in the technologies, and constant doubts about the technologies [3]. Rogers [23] developed the concept as such and argued that consumers go through five stages; knowledge, persuasion, decision, execution, and checking procedures on whether or not to accept an innovative technology or product and show positive responses or skeptical responses regarding whether to accept it or not. There is also a theory of conceptualized resistance to

innovative technology as a concept different from the theory of Sheth [1] and Rogers [23]. That is, the theory perceived resistance to innovative technology as a process that comes from the attitude of the group that accepts the innovative technology rather than a concept opposite to the phenomenon of diffusion of innovative technology when accepted [1]. Most consumers have repulsion against innovative changes because they tend to maintain the current status. Therefore, when innovative new technologies or products are released, if they are resistant to innovation, it is difficult for them to settle sustainably [24].

In another aspect, the authors of [25] define resistance to innovative technology as all behaviors to maintain the current state in the face of pressure to change the current state. On the other hand, Ram sees resistance to innovative technology as an attitude variable and defines it as the degree to which a person feels threatened by changes. Recently, with regard to resistance to innovative technology model, the current study trend is to understand resistance to innovative technology as a concept of attitude rather than simply seeing it as a concept opposite to acceptance and diffusion, and to view that acceptance occurs when resistance to innovative technology has been overcome. This is a very important result of sustainable service delivery.

*2.3. Acceptance Intention*

In general, acceptance intention is a psychological decision made when an innovative product or service is felt to be convenient, easy to use, and useful, and refers to the degree of belief that the accepter can use the innovative product or system with little effort [26]. In addition, the acceptance intention as such can be defined as the consumer's intention to continue to accept the product [23]. Since humans must have an intention first to perform an action, they must have a certain intention to perform an action, and no action is performed without any intention [23]. In this respect, the easier the use of ICT is perceived, and the more useful it is perceived, the more positively attitudes and intentions toward actual use will change and this will result in an increase in the use of ICT. Acceptance intention is regarded as the starting point of actual use, and it becomes a direct determinant of the use of information technology. Therefore, the inclusion of acceptance intention increases the predictive power for actual use compared to no inclusion.

Consumer behavioral intention is presumed to contain motivating factors that affect actual behavior, and attitudes toward behavior are expressed as the degree of preference related to behavior evaluation. Beliefs refer to expectations related to the performance of certain behaviors to achieve outcomes, and evaluation refers to evaluations related to achieving desired outcomes. Subjective norms are defined as perceived social pressures related to performing behaviors and normative beliefs refer to beliefs related to the behavioral expectations of a meaningful reference group. In addition, adaptation motives mean the motivation to pursue the consent of the reference object. While the TRA is pointed out as having the shortcoming of using the abstract concepts of beliefs and evaluations as factors affecting attitude and having no basis for external factors, the TAM is based on the TRA. While the TRA is intended to explain general human behavior, the TAM explains information technology acceptance behavior [11].

**3. Research Design**

*3.1. Research Model and Hypotheses*

In this work, we set the characteristics of unmanned order payment service innovation technology as an independent variable, and set resistance and acceptance intent for unmanned order payment service as a dependent variable. The characteristics of innovative technology for unmanned order payment services consist of a total of three factors: usability, ease of use, and risk that users perceive for unmanned order payment services. It consisted of a single factor for users' resistance to and acceptance of innovative technologies in unmanned order payment services. The model of this study is shown in Figure 1.

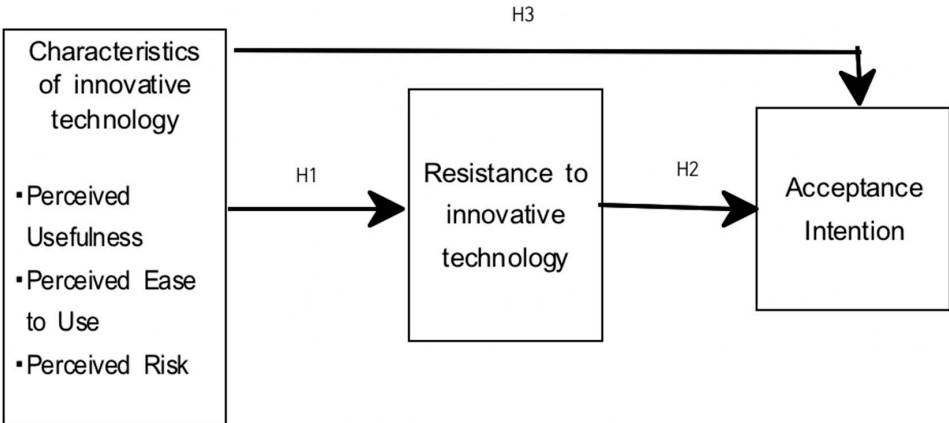

**Figure 1.** Research model.

Shin's [10] work analyzed the resistance to this innovation in the acceptance process of wrist-type wearable devices, showing that perceived usefulness negatively affects innovative resistance, and the complexity of wrist-type wearable devices positively affects innovative resistance. Furthermore, Yoon and Kim [4], who empirically analyzed the resistance to these innovations in the e-book acceptance process, reported that the higher the perceived usefulness, the lower the resistance to e-book acceptance. Additionally, according to a study by Lim et al. (2015) [18], the higher the usability of smartphones, the lower the resistance to smartphone innovation, which can provide sustainable services. Based on this prior study, it can be inferred that the characteristics of innovative technology in unmanned order settlement services will also affect users' resistance to innovative technologies, and therefore the following hypotheses are drawn.

**Hypothesis 1 (H1).** *The characteristics of innovative technology in unmanned order payment services will affect users' resistance to innovative technology.*

**Hypothesis 1 (H1a).** *The perceived usefulness of unmanned order settlement services will have a negative effect on users' resistance to innovative technology.*

**Hypothesis 1 (H1b).** *The perceived ease of use of unmanned order payment services will have a negative effect on users' resistance to innovative technology.*

**Hypothesis 1 (H1c).** *The perceived risk of unmanned order payment services will have a positive effect on users' resistance to innovative technologies.*

Resistance to new innovations and acceptance are not entirely separate. First, it goes through the stage of resistance, then the stage of acceptance. In addition, the higher the resistance to innovative technologies, the lower the acceptance, and conversely, the lower the resistance to innovative technologies, the higher the acceptance. The lower the resistance, the more likely it is to be a sustainable service. A study by Lim et al. [27] reported a negative impact on the acceptance intent of the acceptors' resistance to innovative techniques. In addition, several studies, including Bae [28] and Ram [1] have reported that resistance to innovative techniques by inmates has a negative impact on their acceptance intentions. Ultimately, these findings suggest the need to lower the resistance of inmates to accept and spread innovative technologies. Based on the above prior research, it can be inferred that resistance to innovative technologies in unmanned order payment services will also have a negative impact on users' intention to accept them, thus deriving the following hypothesis.

**Hypothesis 2 (H2).** *User resistance to unmanned payment service innovations can adversely affect sustainable acceptance intentions.*

Jang's [29] study showed that the usefulness of autonomous driving systems has a positive impact on the acceptance intent of the inmates, and perceived ease of use reported a negative impact on the acceptance intent of the inmates. Furthermore, Shin's [30] study also suggested that perceived usefulness of innovative technologies, such as cloud computing services in the information age, has a positive impact on the acceptance intent of inmates. In addition, Choi [31], who empirically analyzed the relationship between the characteristics of innovative technologies and their acceptance intentions, also found that perceived usability and perceived usability of innovative technologies of digital convergence have a positive effect on the acceptance intent of inmates.

Based on the above prior research, it can be inferred that the innovative technology characteristics of unmanned order settlement services will also affect users' intention to accept the innovative technology, thus deriving the following hypothesis.

**Hypothesis 3 (H3).** *Characteristics of innovative technology of unmanned order payment services will affect users' acceptance intention.*

**Hypothesis 3 (H3a).** *The perceived usefulness of unmanned order settlement services will have a positive effect on users' acceptance intention.*

**Hypothesis 3 (H3b).** *Perceived ease of use of unmanned order settlement services will have a positive effect on users' acceptance intention.*

**Hypothesis 3 (H3c).** *Perceived risk of unmanned order settlement services will have a negative effect on users' acceptance intention.*

*3.2. Operational Definition of Research Variables*

3.2.1. Characteristics of Innovative Technology

In this study, innovative characteristics variables for innovative technology in unmanned order payment services are composed of three subfactors: usefulness, ease of use, and risk that users perceive based on Ram and Sheth [24], Kim [32], Park [33], and Kim [34].

The measurement of all questionnaires was measured on the Likert 5-point scale, meaning that the higher the score, the greater the level of usefulness, ease of use, and risk awareness of unmanned order payment service innovation. In particular, recognized risk variables for unmanned order payment service innovation are based on studies by Kim [34] and are not real risks for unmanned order payment service innovation. Measurements of all questions were taken on a Likert 5-point scale, which means that the higher the score, the higher the risk that consumers perceive innovative technologies in unsustainable payment services.

3.2.2. Resistance to Innovative Technology

The user's resistance to innovative technology is not the opposite of acceptance, but the degree to which the user will engage in acceptance and actual use behavior through the user's resistance to innovation [1]. In this work, users' resistance to innovative technologies in unmanned order payment services is defined as resistance to new order and payment methods by insisting on orders and payments through existing employees. In this work, users' resistance to innovative technologies in unmanned order payment services is constructed by a single factor based on the studies of Ram [1], Ram and Sheth [24] and Kim [7]. The measurement of all questionnaires was measured on the Likert 5-point scale, meaning that the higher the score, the higher the user's resistance to innovative technologies in unmanned order payment services.

### 3.2.3. Acceptance Intention

In this work, acceptance variables for innovative techniques in unmanned order payment services were constructed by a single factorization based on studies by Venkatesh and Davis [26] and Yoon [35]. The measurement of all questionnaires was measured on the Likert 5-point scale, meaning that the higher the score, the higher the acceptance intention of unmanned order payment service innovation technology.

### 3.3. Data Processing

The survey data collected for this study were analyzed using the SPSS 26.0 and AMOS 26 statistics programs as follows.

First, frequency and percentage were calculated to identify the general characteristics of the surveyed people.

Second, exploratory factor analysis (EFA) and confirmative factor analysis (CFA) were carried out to verify the feasibility of research variable measurement items such as innovative characteristics, innovative resistance, and acceptance intent of unmanned order settlement services. In order to verify reliability, the Cronbach's coefficient was calculated.

Third, structural equation model (SEM) analysis was conducted to verify the relationship between innovative characteristics, innovative resistance, and acceptance of unmanned order payment services, and the medium effect of innovative resistance was verified through bootstrapping. The significance level of all statistical analysis and hypothesis validation was performed at 0.05.

## 4. Results

### 4.1. Demographic Characteristics of the Subjects Surveyed

The subjects of this study are Korean, and adult men and women in their 20s and older living in Seoul. The survey was conducted over about four weeks from the first week to the fourth week of May 2021, and the response results were collected through a mobile survey. The demographic characteristics of the consumers surveyed are shown in Table 1. Gender consisted of 185 men (61.1%) and 118 women (38.9%), and age was 82 in their 20s (27.15%), 165 in their 30s (54.5%), 22 in their 40s (7.35%), and 34 in their 50s and older (11.2%).

**Table 1.** Demographic characteristics of the subjects surveyed.

| | **Variable** | **N** | **%** |
|---|---|---|---|
| Sex | Man | 185 | 61.1 |
| | Woman | 118 | 38.9 |
| Age | 20s | 82 | 27.1 |
| | 30s | 165 | 54.5 |
| | 40s | 22 | 7.3 |
| | 50s and older | 34 | 11.2 |
| Job | Office workers/professionals | 225 | 74.3 |
| | Self-employed/businesses | 36 | 11.9 |
| | Student | 38 | 12.5 |
| | Housewife | 4 | 1.3 |
| Degree of use of unmanned order payment services | Often | 203 | 67.0 |
| | Every time | 82 | 27.1 |
| | Total | 303 | 100.0 |

The jobs were office workers/professionals 225 (74.3%) followed by 38 students (12.5%), 36 self-employed/businesses (11.9%), and 4 housewives(1.3%). The degree of use

of unmanned order payment services at restaurants and fast-food restaurants was 203 (27.1%) and 82 (27.1%) always used.

*4.2. Verification of Validity and Reliability*

4.2.1. Exploratory Factor Analysis and Verification of Reliability

Exploratory factor analysis was conducted to verify the concept validity of the measurements of innovative characteristics of unmanned order payment services used in this study. Principal component analysis was conducted in factor analysis, and the rotation of the factor was varimax. In this study, questions with a factor loading of 0.5 or less and factor loading were removed when factor loading was allocated to two or more factors. In this work, the objective was to refine the measurement items and achieve conceptual validity by applying these criteria. Next, Cronbach's $\alpha$ value was calculated for reliability verification, which indicates the internal consistency between items that constitute the factors extracted through factor analysis. In general, a value of Cronbach's $\alpha$ value above 60 can be considered reliable.

First of all, the results of exploratory factor analysis and reliability analysis on the innovative characteristics measurement items of unmanned order payment services used in this study are shown in Table 2. Factor analysis removed three measurement items with low factor loading or high load on factors with different research concepts. The Kaiser–Meyer–Olkin (KMO) value for determining sample fit was 0.871. Bartlett's sphericality verification results showed significant values, and therefore the collected data and measurements were suitable for performing factor analysis. The factor analysis showed that three factors were extracted, with a total explanation of variance of 69.248%. Factor 1 is a 'perceived ease of use' factor, dispersive explanation is 27.339%, Factor 2 is a 'perceived usefulness' factor that is 23.567%, and Factor 3 is a 'perceived risk' factor that is 18.342%, confirming concept validity. In this study, technical characteristics were presented as three factors: perceived usefulness, perceived ease of use, and perceived risk. Other researchers' studies related to TAM also suggested three factors: perceived usefulness, perceived ease of use, and perceived risk. Shen et al. (2015) verified the effect of Digital Textbook Learning System characteristics on the intention to use by applying TAM to Digital Textbook Learning System for Chinese consumers [36]. Through factor analysis, they presented the characteristics of the Digital Textbook Learning System as three factors: perceived usefulness, perceived ease of use, and perceived security risk. Thar and Riyadh (2020) verified the effect of E-Filling technology characteristics on the intention to use by applying TAM to E-Filling technology to Indonesian consumers [37]. Through factor analysis, the researchers suggested the characteristics of E-Filing technology as three factors: perceived usefulness, perceived ease of use, and perceived risk.

Next, the reliability of the items that make up the innovative characteristics factors of unmanned order payment services was verified as Cronbach's $\alpha$ value and were perceived ease of use 0.881, perceived usefulness 0.841, perceived risk 0.859, respectively, consisting of all factors internally consistent.

The results of the exploratory factor analysis and reliability analysis on the items of innovation resistance and acceptance of unmanned order settlement services are as shown in Table 3. As a result of exploratory factor analysis, KMO value, 0.864, Bartlett's sphericity, approximated $\chi^2 = 1575.4420$ ($df = 28$, $p < 0.001$) showed that the measurement items were therefore suitable for performing factor analysis. Factor analysis showed that no items were removed, and two factors were extracted. Total distributed explanation power was 73.666%. Factor 1 was 'resistance to innovative technology' to unmanned order payment services, variance was 42.989% and factor 2 was 'acceptance intention' for unmanned order payment services, with variance being 30.677%, respectively, and construct validity was confirmed. Next, after verifying the reliability of resistance to innovative technology and acceptance intention for unmanned order payment services, the Cronbach's $\alpha$ value of the innovation resistance was 0.881, and the Cronbach's $\alpha$ value of acceptance was 0.877,

respectively. Therefore, it consisted of items with internal consistency and the reliability was confirmed.

**Table 2.** Factor analysis and reliability verification of characteristics of innovative technology.

| Factor | Item | Factor Loading | | | Cronbach's $\alpha$ |
| --- | --- | --- | --- | --- | --- |
| | | Factor 1 | Factor 2 | Factor 3 | Cronbach's $\alpha$ |
| Perceived Usefulness | * Learning for the use of unmanned order payment services will be difficult (UA2) | 0.863 | 0.118 | −0.139 | 0.881 |
| Perceived Usefulness | * Manipulation for the use of unmanned order payment services will be complicated (UA3) | 0.847 | 0.167 | −0.188 | 0.881 |
| Perceived Usefulness | * Procedures for using unmanned order payment services will be difficult (UA5) | 0.765 | 0.205 | −0.128 | 0.881 |
| Perceived Usefulness | * Unmanned order payment service will be inconvenient when used (UA 1) | 0.720 | 0.066 | −0.246 | 0.881 |
| Perceived Usefulness | * It will be difficult to identify devices and functions when using unmanned order payment services (UA4) | 0.710 | 0.232 | −0.226 | 0.881 |
| Perceived Ease of Use | The unmanned order payment service will be convenient because unlike existing store employees, there is no need for words when ordering (PU2) | 0.120 | 0.847 | −0.024 | 0.841 |
| Perceived Ease of Use | Unmanned order payment service will be more useful than orders through existing store staff as it can provide optimal cibis environment such as menu and price (PU1) | −0.144 | 0.790 | −0.104 | 0.841 |
| Perceived Ease of Use | Unmanned order payment services will be more productive because they do not require existing store staff (PU6) | 0.270 | 0.750 | −0.095 | 0.841 |
| Perceived Ease of Use | Unmanned order payment services will save money on orders made through existing store staff (PU5) | 0.276 | 0.736 | −0.061 | 0.841 |
| Perceived Ease of Use | The unmanned order payment service will be time-efficient as it allows faster orders from existing store staff (PU3) | 0.384 | 0.674 | 0.022 | 0.841 |
| Perceived Risk | Unmanned order payment services will be more at risk of personal information leakage than orders made to existing store staff (PR1) | −0.142 | −0.146 | 0.856 | 0.859 |
| Perceived Risk | Unmanned order payment services would pose a risk to hacking (PR 2) | −0.293 | −0.011 | 0.856 | 0.859 |
| Perceived Risk | Unmanned order payment services will have problems during operation (PR3) | −0.217 | −0.033 | 0.844 | 0.859 |
| | Eigen Value | 3.554 | 3.064 | 2.384 | |
| | Variance (%) | 27.339 | 23.567 | 18.342 | |
| | Total Variance (%) | 27.339 | 50.906 | 69.248 | |
| KMO (Kaiser–Meyer–Olkin) = 0.871, Bartlett's Sphericity: Approximated $\chi^2$ = 2247.479 (*df* = 78, *p* = 0.000) | | | | | |

*: Reversed calculated.

**Table 3.** Factor analysis and reliability verification of resistance to innovative technology and acceptance intention.

| Factor | Item | Factor Loading | | Cronbach's $\alpha$ |
|---|---|---|---|---|
| | | Factor 1 | Factor 2 | |
| Resistance to Innovative Technology | I have resistance to the use of unmanned order payment services in restaurants, fast food restaurants, cafes, etc. (IR1) | 0.749 | −0.202 | 0.881 |
| Resistance to Innovative Technology | I do not want to use unmanned order payment services in restaurants, fast food restaurants, cafes, etc. (IR3) | 0.801 | −0.296 | 0.881 |
| Resistance to Innovative Technology | I have anxiety about the use of unmanned order payment services in restaurants, fast food restaurants, cafes, etc. (IR2) | 0.867 | −0.218 | 0.881 |
| Resistance to Innovative Technology | I am willing to oppose the use of unmanned order payment services in restaurants, fast food restaurants, cafes, etc. (IR4) | 0.810 | −0.261 | 0.881 |
| Resistance to Innovative Technology | I am not interested in using unmanned order payment services at restaurants, fast food restaurants, cafes, etc. (IR 5) | 0.685 | −0.312 | 0.881 |
| Acceptance Intention | It is positive about the use of unmanned order payment services in restaurants, fast food restaurants, cafes, etc. (AI1) | −0.394 | 0.739 | 0.877 |
| Acceptance Intention | I am willing to use unmanned order payment services at restaurants, fast food restaurants, cafes, etc. (AI 2) | −0.171 | 0.903 | 0.877 |
| Acceptance Intention | I intend to recommend unmanned order payment service services to acquaintances around me (AI3) | −0.297 | 0.866 | 0.877 |
| | Eigen Value | 3.439 | 2.454 | |
| | Variance (%) | 42.989 | 30.677 | |
| | Total Variance (%) | 42.989 | 73.666 | |
| KMO (Kaiser–Meyer–Olkin) = 0.864, Bartlett's Sphericity: Approximated $\chi^2$ = 1575.442 (*df* = 28, *p* = 0.000) | | | | |

### 4.2.2. Confirmatory Factor Analysis

Confirmative factor analysis was conducted on the measurement model to verify the convergence and validity of research variables such as perceived usability, perceived ease of use, perceived risk factors, and innovation resistance and acceptance of unmanned order payment services. To assess the fit of a measurement model, it is important to have an analytical basis without being sensitive to the size of the sample, and to select an appropriate fit index considering the simplicity of the mode. In this work, we explore the model's fit through goodness-of-fit indices such as $\chi^2$ value, SRMR (Standardized Root Mean Square Residual), TLI (Tucker Lewis Index), CFI (Comparative Fit Index), and RMSA (Root Mean Square Error or Approach). Typically, $\chi^2$ value is suitable on $p > 0.05$, other goodness-of-fit indices are given priority because $\chi^2$ value is sensitive to the number of samples. Generally, TLI and CFI are evaluated as good fit if 0.90 or higher, and SRMR is evaluated as good fit if 0.08 or less. For RMSEA where confidence intervals are presented, 0.05 or less is rated as good fit, and 0.08 or less as good fit, and 0.10 or less is rated as moderate fit. The revised index was reviewed through confirmatory factor analysis, and item No. 4 of 'perceived ease of use' was further removed. Looking at the fit of the measurement model given in Table 4, $\chi^2$ = 426.108 (*df* = 160, $p < 0.001$), SRMR = 0.047, TLI = 0.923, CFI = 0.935, RMSEA (90% CI) = 0.059 (0.051~0.067), and so on, showing good fit, indicating that the measurement model is suitable. In addition, the factor loading of all measurement variables for potential variables such as perceived usefulness, perceived usability, perceived risk factors, and willingness to accept unmanned order payment

services was statistically significant ($p < 0.001$), the standardized factor loading was all higher than 0.50, and no theoretical negative distribution was found.

**Table 4.** Confirmative factor analysis.

| Variable | | Non-Standardized Factor Loading | Standard Error | Standardized Factor Loading | Variance of Error | *t* | Construct Reliability (CR) | Average Variance Extracted (AVE) |
|---|---|---|---|---|---|---|---|---|
| Perceived Usefulness | PU 1 | 1.000 | - | 0.617 | 0.344 | - | 0.865 | 0.562 |
| Perceived Usefulness | PU 2 | 1.625 | 0.156 | 0.771 | 0.379 | 10.428 *** | 0.865 | 0.562 |
| Perceived Usefulness | PU 3 | 1.733 | 0.173 | 0.726 | 0.571 | 10.000 *** | 0.865 | 0.562 |
| Perceived Usefulness | PU 5 | 1.420 | 0.140 | 0.742 | 0.347 | 10.162 *** | 0.865 | 0.562 |
| Perceived Usefulness | PU 6 | 1.611 | 0.156 | 0.759 | 0.404 | 10.315 *** | 0.865 | 0.562 |
| Perceived Ease of Use | PUA1 | 1.000 | - | 0.729 | 0.573 | - | 0.857 | 0.602 |
| Perceived Ease of Use | PUA2 | 1.182 | 0.083 | 0.867 | 0.300 | 14.317 *** | 0.857 | 0.602 |
| Perceived Ease of Use | PUA3 | 1.013 | 0.071 | 0.858 | 0.239 | 14.208 *** | 0.857 | 0.602 |
| Perceived Ease of Use | PUA5 | 0.896 | 0.077 | 0.698 | 0.547 | 11.628 *** | 0.857 | 0.602 |
| Perceived Risk | PR 1 | 1.000 | - | 0.765 | 0.483 | - | 0.848 | 0.652 |
| Perceived Risk | PR 2 | 1.201 | 0.081 | 0.900 | 0.229 | 14.858 *** | 0.848 | 0.652 |
| Perceived Risk | PR 3 | 0.961 | 0.069 | 0.794 | 0.369 | 13.858 *** | 0.848 | 0.652 |
| Resistance to Innovative Technology | IR 1 | 1.000 | - | 0.716 | 0.436 | - | 0.905 | 0.658 |
| Resistance to Innovative Technology | IR 2 | 1.049 | 0.076 | 0.831 | 0.227 | 13.756 *** | 0.905 | 0.658 |
| Resistance to Innovative Technology | IR 3 | 1.117 | 0.078 | 0.861 | 0.199 | 14.227 *** | 0.905 | 0.658 |
| Resistance to Innovative Technology | IR 4 | 1.001 | 0.076 | 0.796 | 0.265 | 13.207 *** | 0.905 | 0.658 |
| Resistance to Innovative Technology | IR 5 | 0.941 | 0.083 | 0.687 | 0.455 | 11.404 *** | 0.905 | 0.658 |
| Acceptance Intention | AI 1 | 1.000 | - | 0.888 | 0.145 | - | 0.911 | 0.774 |
| Acceptance Intention | AI 2 | 0.801 | 0.050 | 0.772 | 0.234 | 16.013 *** | 0.911 | 0.774 |
| Acceptance Intention | AI 3 | 1.035 | 0.057 | 0.843 | 0.233 | 18.257 *** | 0.911 | 0.774 |
| $\chi^2 = 426.108$ ($df = 160$, $p = 0.000$), SRMR = 0.047, TLI = 0.923, CFI = 0.935, RMSEA (90% CI) = 0.059 (0.051~0.067) | | | | | | | | |

*** $p < 0.001$.

Next, to examine the convergent validity of the latent variables, we examine the construct reliability (CR) and the average variance extracted value (AVE). First, convergent validity refers to the degree of correlation between two or more measurements for one latent variable, and is generally considered to have a construct reliability of 0.70 or higher,

and an average variance extracted value of 0.50 or higher. As shown in Table 4, construct reliability of all latent variables is over 70, perceived usability (0.865), perceived usability (0.857), perceived risk (0.848) and perceived risk (0.905) for unmanned order payment services (0.911). The average variance extracted values of all latent variables were also found to be greater than 0.50, perceived usefulness (0.562), perceived ease of use (0.602), perceived risk (0.652) and resistance to innovative technology (0.658), acceptance intention (0.774). Therefore, convergent validity was identified.

Finally, we looked at the discrimination between the latent variables. Discriminant validity indicates how different one latent variable actually is from the other, and the most conservative evaluation method considers it to be discriminative if each latent variable's AVE value is greater than the square of the correlation coefficient of the two latent variables. By comparing the squared values of the correlation coefficients and AVE values given in Table 5, the discrimination between the latent variables was found to be lower than the squared value ($-0.526$) of the correlation coefficients ($-0.725$) between the highest correlation unmanned order payment services.

**Table 5.** Correlations between the research variables.

| Variable | Characteristics of Innovative Technology | | | Resistance to Innovative Technology | Acceptance Intention |
|---|---|---|---|---|---|
| | Perceived Usefulness | Perceived Ease of Use | Perceived Risk | | |
| Perceived Usefulness | 0.562 | | | | |
| Perceived Ease of Use | 0.465 *** | 0.602 | | | |
| Perceived Risk | $-0.212$ ** | $-0.502$ *** | 0.652 | | |
| Resistance to Innovative Technology | $-0.599$ *** | $-0.530$ *** | 0.450 *** | 0.658 | |
| Acceptance Intention | 0.700 *** | 0.382 *** | $-0.297$ *** | $-0.725$ *** | 0.774 |

** $p < 0.01$, *** $p < 0.001$, The values on diagonal line mean AVE.

Looking at the correlations among the variables, perceived usefulness, perceived ease of use, and perceived risk of innovation in unmanned order payment services showed significant negative correlations with resistance to innovative technology for unmanned order payment services.

It showed significant positive correlation with acceptance intention, and perceived risk factors showed a significant positive correlation with resistance to innovative technology for unmanned order payment services. There was a significant negative correlation with acceptance intention. There was a significant negative correlation between resistance to innovative technology and acceptance intention of unmanned order payment services.

### 4.3. Verification of Research Hypotheses

To verify the research hypothesis, a structural equation model analysis was performed using AMOS 26.0, and the parameter estimation method used Maximum Likelihood (ML). First of all, fitness of the research model was analyzed. As shown in Table 6, the results showed that $\chi^2 = 426.108$ ($df = 160$, $p < 0.001$), SRMR = 0.047, TLI = 0.923, CFI = 0.935, RMSEA (90% CI) = 0.059 (0.051~0.067). Therefore, the fit of the research model was excellent, and it was analyzed that it was fit to accept the research results.

The verification results of research hypotheses 1, 2 and 3 to verify the effect of perceived usefulness, perceived ease of use, and perceived risk on resistance to innovative technology and acceptance intention, and the effect of resistance to innovative technology on acceptance intention are shown in Table 7.

**Table 6.** Fitness of the research model.

| $\chi^2$ | *df* | *p* | SRMR | TLI | CFI | RMSEA (90% CI) |
|---|---|---|---|---|---|---|
| 426.108 | 160 | 0.000 | 0.047 | 0.923 | 0.935 | 0.059 (0.051~0.067) |

**Table 7.** Verification result of research hypotheses 1, 2, 3.

| Path | | Non-Standardized Coefficient | Standard Error | Standardized Coefficient | *t(C.R)* | *p* |
|---|---|---|---|---|---|---|
| Perceived Usefulness | Resistance | −0.674 | 0.109 | −0.457 | −6.187 | 0.000 |
| Perceived Ease of Use | Resistance | −0.158 | 0.059 | −0.188 | −2.684 | 0.007 |
| Perceived Risk | Resistance | 0.213 | 0.052 | 0.259 | 4.082 | 0.000 |
| Acceptance Intention | Acceptance Intention | −0.543 | 0.082 | −0.502 | −6.645 | 0.000 |
| Perceived Usefulness | Acceptance Intention | 0.703 | 0.118 | 0.441 | 5.948 | 0.000 |
| Perceived Ease of Use | Acceptance Intention | −0.094 | 0.058 | −0.104 | −1.638 | 0.102 |
| Perceived Risk | Acceptance Intention | −0.026 | 0.052 | −0.030 | −0.511 | 0.610 |

Looking at the verification results of the research hypothesis 1 (Figure 2), which predicted that characteristics of innovative technology in unmanned order payment services will affect users' resistance to innovative technology, the perceived usefulness of characteristics of innovative technology of unmanned order payment services (standardized path coefficient t = −0.674, t = −6.187, *p* < 0.001), perceived ease of use (standardized path coefficient = −0.188, t = −2.684, *p* < 0.01) was shown to have a significant negative effect on resistance to innovative technology, and perceived risk (standardized path coefficient = 0.259, t = 4.082, *p* < 0.001) have been shown to have a significant positive effect on resistance to innovative technology. These results show that the higher consumers perceive the usefulness and ease of use of unmanned order payment services, the lower the resistance to unmanned order payment services, and the higher the risk, the higher the resistance. Thus, research hypotheses 1-1, 1-2 and 1-3 were adopted.

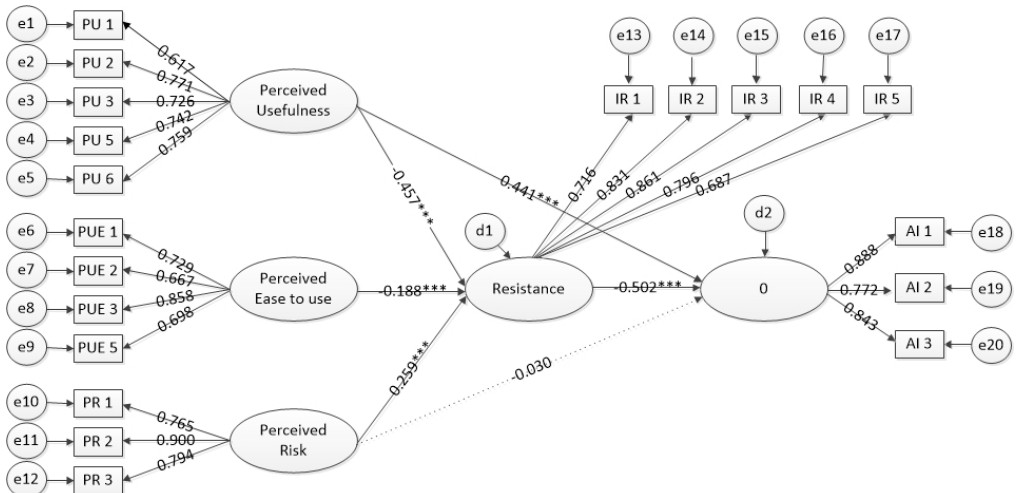

**Figure 2.** Verification result of research hypotheses 1, 2, 3. * *p* < 0.05, ** *p* < 0.01, *** *p* < 0.001.

Next, looking at the verification results of research hypothesis 2, which predicts that resistance to unmanned order payment services will affect acceptance intention, resistance

to unmanned order payment services has a negative effect on acceptance intention (standardized path coefficient = −0.502, t = −6.645, $p < 0.001$). These results mean that the lower the consumer's resistance to unmanned order payment services, the higher the acceptance intention of unmanned order payment services. Therefore, research hypothesis 2 of the study was adopted.

Looking at the results of the verification of the research hypothesis 3, which predicted that the characteristics of innovative technology of unmanned order payment services would affect the acceptance intention, the perceived usefulness of unmanned order payment services directly affects the acceptance intention (standardized path coefficient = 0.441, t = 5.9481, $p < 0.001$) and was shown to have a significant positive effect on acceptance intention. However, perceived ease of use and perceived risk did not have a significant effect directly on acceptance intention. These results indicate that the higher consumers perceive the usefulness of unmanned order payment services, the higher the acceptance intention of unmanned order payment services, so the perceived usefulness among characteristics of innovative technology of unmanned order payment services is a major predictor. Thus, research hypothesis 3-1 was adopted, but 3-2 and 3-3 were rejected.

Next, bootstrapping was performed on the indirect effects of the path between the characteristics of innovative technology and acceptance intention to verify the research hypothesis 4, which predicted the mediating effects of resistance to innovative technology in the relationship between characteristics of innovative technology and acceptance intentions of unmanned order payment services. Bootstrapping is a method of estimating the distribution of parameters based on sample data without knowing the distribution of the population. When the 95% confidence interval (CI) does not contain zero, it is considered significant at the significance level of 0.05. The results of the bootstrapping analysis are shown in Table 8.

**Table 8.** Verification result of research hypothesis 4.

| Path | | | Indirect Effect (Bootstrapping) | | | | |
|---|---|---|---|---|---|---|---|
| Path | | | Non-Standardized Coefficient | Standard Error | Standardized Coefficient | 95% CI | *p* |
| Perceived Usefulness | Resistance | Acceptance Intention | 0.366 | 0.085 | 0.229 | (0.224~0.563) | 0.001 |
| Perceived Ease of Use | Resistance | Acceptance Intention | 0.086 | 0.039 | 0.094 | (0.020~0.173) | 0.009 |
| Perceived Risk | Resistance | Acceptance Intention | −0.116 | 0.037 | −0.130 | (−0.206~−0.054) | 0.000 |

Bootstrapping sampling (N = 2000).

As a result of verifying the medium effect of innovation resistance in the relationship between innovation characteristics of unmanned order payment services and acceptance intentions, the perceived usefulness of unmanned order payment services → resistance → acceptance intention path (non-standardized path coefficient = 0.366, 95% CI: 0.224~0.563, $p < 0.01$), perceived ease of use → resistance → acceptance intention path (non-standardized path coefficient = 0.086, 95% CI: 0.020~0.173, $p < 0.01$), perceived risk → resistance → acceptance intention path (non-standardized path coefficient = −0.116, 95% CI: −0.206~−0.054) showed that all did not include zero in the 95% confidence interval. These results confirm that perceived usefulness, perceived ease of use, and perceived risk among the characteristics of innovative technology of unmanned order payment services affect acceptance intention through the mediating of resistance to innovative technology. Therefore, research hypotheses 4-1, 4-2, and 4-3 were all adopted. Combining the above results, perceived ease of use and perceived risk of unmanned order payment services did not directly affect the acceptance intention, but only through the mediating of resistance to innovative technology, and the full-mediated effect was shown. The perceived usefulness not only directly affects the acceptance intention, but also affects the acceptance intention through the mediating of resistance to innovative technology, and the partial mediating effect was confirmed.

## 5. Discussion

This study was designed to empirically verify innovative technology characteristic factors for the sustainable use and distribution of consumer's innovative payment services, a new payment method in the era of the Fourth Industrial Revolution. This study sought to establish a structural relationship between characteristics of innovative technology of unmanned order payment services (perceived usefulness, perceived ease of use, perceived risk) and resistance to innovative technology and acceptance intent. A survey was conducted on consumers experiencing unmanned order payment services residing in Seoul, and the results of the hypothesis verification of this study are as follows.

First, verification of the impact of innovative characteristics on innovative resistance of unmanned order payment services showed that perceived usefulness and ease of use had a significant negative effect on innovative resistance, and perceived risk had a significant positive effect on resistance. Shin's [21] study shows that the relative advantage has a significant negative effect on the resistance, and complexity has a significant positive effect on the resistance. Relative benefits correspond to perceived usefulness of this study, and complexity corresponds to perceived ease of use, indicating that these results tend to be partially consistent with the findings of this study and support them overall. Furthermore, the results of this study show that the higher the perceived usefulness of e-books, the lower the resistance, and the higher the perceived usefulness for smart phones, the lower the innovation resistance for smart phones.

Second, after verifying the effect of resistance to unmanned order payment services on consumer's acceptance intention, consumers' resistance to unmanned order payment services has a significant negative effect on their acceptance intention. Rogers [23] insisted that when resistance to innovative technology is relaxed, inmates can accept it, and if this resistance is stronger than any level, the timing of acceptance will be delayed or not accepted at all, which is in line with the findings of the study. As a result, minimizing the resistance of the acceptor has a positive impact on sustainable service demand. Furthermore, the results of Lim et al. [18], Bae [19] and Ram [1] show that the resistance to the innovative technology has a negative effect on the acceptance intention, therefore these results are supporting the results of this study.

Third, the effect of characteristics of innovative technology such as unmanned order payment services on the acceptance intention was verified, and the perceived usefulness of unmanned order payment services directly affected the acceptance intention, but not the perceived ease of use and risk. Studies by Jang [20] showed that the relative benefits of autonomous vehicle systems had a significant positive effect on the acceptance intention, and complexity factors had a negative effect on the acceptance intention, but the perceived risk did not have a significant effect on the acceptance intention. Relative benefits correspond to perceived usefulness of this work, and complexity corresponds to perceived ease of use, so these results can be seen as partially consistent with the results of this study. Shin [21] also reported that perceived usefulness of innovative technologies such as cloud computing services in the information age has a positive effect on the acceptance intention of cloud computing services. This can be seen as a tendency to correspond with the findings of this study. Furthermore, the findings of Choi [6], who reported that perceived usefulness and perceived ease of use of innovation in digital convergence have a positive effect on the acceptance intention, also show a tendency to align with this work, and thus support the findings of this study. That is, it is advantageous in terms of sustainability to minimize the complexity of unmanned order payment services and to increase the usefulness of recognition of innovative technologies by governments and businesses.

Fourth, the relationship between characteristics of innovative technology and acceptance intention of unmanned order payment service, the resistance to innovative technology has a mediated effect. These results mean that the innovative characteristics of unmanned order payment services, that is, the usefulness, ease of use and risk of unmanned order payment services perceived by consumers, affect consumer's resistance and ultimately their intention to accept them.

This study is meaningful in that it derived factors affecting the resistance and acceptance of unmanned order payment services by applying them to unmanned order payment services, including the variable of resistance to innovative technology in the course of consumer acceptance in the existing technology acceptance model (TAM). The methodology for analyzing these factors is critical for innovative technologies to become a sustainable service industry. This study expects to contribute to the sustainable industry development of unmanned payment systems.

## 6. Limitation and Further Study

This study empirically investigated the relationship between the characteristics of unmanned order payment services newly introduced and distributed in many stores in Korea and consumers' resistance and acceptance intention to innovative technologies. When new innovative technologies and services emerge, consumers show resistance and acceptance of new innovations. In order to reduce resistance in this process, it is necessary to increase the usefulness and ease of use perceived by consumers, and to reduce the perceived risk. On the other hand, for sustainable use, consumers must lower their resistance to innovation and increase their acceptance intention, which is closely related to the usefulness, ease of use, and perceived risk perceived by consumers. Therefore, for sustainable use and diffusion, efforts to increase the usefulness and ease of use perceived by consumers and to lower perceived risk are required.

The study was conducted on only 303 consumers experiencing unmanned order payment services living in Seoul in Korea, so there may be limitations in generalizing the results of this study. Therefore, further research needs to further generalize the results of the study by conducting comprehensive survey studies, including consumers experiencing unmanned payment services outside of Seoul or foreign countries.

**Funding:** This research received no external funding.

**Institutional Review Board Statement:** Not applicable.

**Informed Consent Statement:** Not applicable.

**Data Availability Statement:** Not applicable.

**Conflicts of Interest:** The author declares no conflict of interest.

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
