# Peer review of "The Mediating Role of Resistance to Innovative Technology between the Characteristics of Innovative Technology and Sustainable Use of Innovative Payment Service"

_sustainability, doi:10.3390/su131910984_

Round 1
Reviewer 1 Report
Thank you for giving me the opportunity to review the article „The Mediating role of Resistance to Innovative Technology between the Characteristics of Innovative Technology and Sustainable Use of Innovative Payment Service”. Below my remarks:
- the Author has chosen an interesting and current topic on factors influencing the use of innovative payment services;
- the abstract is a little too long. It should shorten the description of the results and include only the main ones;
- the introduction lacks information on what is included in the various sections of the article;
- the literature review is very poor. The author should study the literature and analyse other studies using TAM for payment services/forms. Analyse which constructs on this issue have been analysed;
- research process prepared by the Author is correct. However, I am very doubtful whether 303 respondents constitute a representative research sample on which to draw conclusions. I think that this survey could be described as preliminary;
- in the section "Demographic Characteristics" there is a definite lack of indication of the country (place) of origin of the respondents. I also could not find information on when and in which form and with which tool the research was conducted;
- The results of the factor analysis are described in great detail. However, I definitely miss the discussion of the results, including the reference to results obtained by other researchers (e.g. from other countries).
- The "Conclusion" and “Limitations and Future Research” sections should be separated from the Discussion section;
- technical corrections are necessary. The article should be adapted to the requirements of the journal:
- References should be properly prepared. E.g. Kalmykova, Y.; Harder, R.; Borgestedt, H.; Svanäng, I. Pathways and Management of Phosphorus in Urban Areas. Ind. Ecol. 2012, 16, 928–939.
Author Response
Please see the attachment.
Reviewer 1 is marked in red, and reviewer 2 to 3 are marked in blue.

Reviewer 2 Report
Dear Authors,
thank you for your contribution and interesting results. However few improvements would enhance the quality. I suggest you to emphasize the purpose of the study. As well it would be great to have more recent research articles discussed in Theoretical background. As well, the discussion part should be presented separately from conclusions. Provide main discussion and insights on research results in Discussion part, and overall conclusions by discussing main aspect of all article.
Author Response

(The authors gave the same response as above.)

Reviewer 3 Report
Thank you for your paper. Your paper has all the requirements for publication. I believe improving the discussion and conclusion section improve this paper. Also, improve the reference section. You do not have any paper for 2020-21. Updated references are required.
Author Response

(The authors gave the same response as above.)

Round 2
Reviewer 1 Report
thank you for taking my comments into account
Reviewer 2 Report
Thank you for the improvements of paper.